# Network-based prediction of the disclosure of ideation about self-harm and suicide in online counseling sessions

Zhongzhi Xu [1,2], Christian S. Chan [3,4✉], Qingpeng Zhang [5], Yucan Xu [2], Lihong He[2], Florence Cheung[2], Jiannan Yang[5], Evangeline Chan[2], Jerry Fung [2], Christy Tsang[2], Joyce Liu[2] & Paul S. F. Yip [2✉]

## Abstract

**Background** In psychological services, the transition to the disclosure of ideation about self-harm and suicide (ISS) is a critical point warranting attention. This study developed and tested a succinct descriptor to predict such transitions in an online synchronous text-based counseling service.

**Method** We analyzed two years' worth of counseling sessions ($N = 49{,}770$) from Open Up, a 24/7 service in Hong Kong. Sessions from Year 1 ($N = 20{,}618$) were used to construct a word affinity network (WAN), which depicts the semantic relationships between words. Sessions from Year 2 ($N = 29{,}152$), including 1168 with explicit ISS, were used to train and test the downstream ISS prediction model. We divided and classified these sessions into ISS blocks (ISSBs), blocks prior to ISSBs (PISSBs), and non-ISS blocks (NISSBs). To detect PISSB, we adopted complex network approaches to examine the distance among different types of blocks in WAN.

**Results** Our analyses find that words within a block tend to form a module in WAN and that network-based distance between modules is a reliable indicator of PISSB. The proposed model yields a c-statistic of 0.79 in identifying PISSB.

**Conclusions** This simple yet robust network-based model could accurately predict the transition point of suicidal ideation prior to its explicit disclosure. It can potentially improve the preparedness and efficiency of help-providers in text-based counseling services for mitigating self-harm and suicide.

**Plain language summary**

In online counseling, the help-provider can often be engaging with several service users simultaneously. Therefore, new tools that could help to alert and assist the help-provider and increase their preparedness for getting further help for service users could be useful. In this study, we developed and tested a new tool that is designed to alert help-providers to the disclosure of self-harm and suicidal thoughts, based on the words that the service user has been typing. The tool is developed on the basis that word usage may have a specific pattern when suicidal thoughts are more likely to occur. We tested our tool using two years' worth of online counseling conversations and we show that our approach can help to predict the confession of suicidal thoughts. As such, we are taking a step forward in helping to improve these counseling services.

[1] School of Public Health, Sun Yat-sen University, Guangzhou, China. [2] Hong Kong Jockey Club Centre for Suicide Research and Prevention, The University of Hong Kong, Hong Kong SAR, China. [3] Department of Psychology, The University of Hong Kong, Hong Kong SAR, China. [4] Department of Psychology and Linguistics, International Christian University, Tokyo, Japan. [5] School of Data Science, City University of Hong Kong, Hong Kong SAR, China. ✉email: shaunlyn@hku.hk; sfpyip@hku.hk

Online psychological services grow rapidly since their inception in the 1990s[1,2]. These services provide an important avenue for preventing self-harm and suicide, which is a major public health challenge and the leading cause of death among youth worldwide.

Self-harm and suicide risk assessment and intervention is especially challenging in the context of online services because of the lack of contextual and behavioral information and the inability to physically intervene the crisis[3]. The anonymity, while a major strength of some online services[4], also adds to the complexity in providing the needed crisis intervention to those at risk. Self-harm and suicide assessment and intervention in online psychological services primarily rely on the explicit prompting and disclosure of the help-seeker[5–9]. In many ways, such explicit forms of disclosure are *postdictors* rather than predictors of the ideation of self-harm and suicide (ISS)[10]. It would be of tremendous theoretical interest and practical use if the shift from non-ISS utterance to the disclosure of ISS can be identified earlier on, especially when risk assessment and intervention are not as readily available or deployable as face-to-face services[10]. This is because, if the transition point from non-ISS language to ISS language can be predicted, help-providers can prepare for the potential disclosure of ISS. Such preparedness includes, for example, deploying their skills to explore ISS as many help-seekers tend to conceal their ISS during psychological treatment[11]. Moreover, in the contexts in which the same text-based help-provider is serving multiple texters simultaneously, the prediction of transition points can serve as an alert system to prompt the provider to prioritize and manage their allocation of attention and time. This would potentially improve the efficiency and effectiveness of text-based services, as one-to-many counseling becomes feasible and safe, and without compromising the users' experiences.

To date, research lending empirical support to the prediction of the transition point to ISS is rare. To our knowledge, there are only three recent studies that have investigated the prediction of ISS in social media[10,12,13]. The authors share our concern that most studies assess text-based messages, which include explicit mentions of ISS, but do not address the *prediction* of ISS. While those studies have demonstrated some promise of predicting ISS, the deep learning algorithms that have been adopted in both studies are inherently *black boxes*; one cannot extract any human-understandable insights from the model as even its designers cannot explain why such an artificial intelligence arrives at a specific decision. In addition, the linguistic patterns between social media posts and counseling sessions are diametrically different as social media posts are self-utterance while counseling consists of messages exchanged between two people. As such, our knowledge about predicting ISS in a text-based online counseling session is still lacking, and a transparent yet robust model for such analysis is needed.

The present study aims to fill the research gap by developing and testing techniques based on complex network theory in large-scale text-based counseling data. To do so, we first construct a word affinity network (WAN) that depicts the semantic relationships between words used in text-based counseling. WAN serves as a substrate on top of which the downstream calculation of network-based distance takes place. We then use the tools of network science[14–16] to develop and test the accuracy of a framework predicting the transition point to ISS.

## Methods

**Approval of ethics**. Users gave their consent to using their text data for research purposes by accepting the Privacy Policy before the commencement of service. The Privacy Policy can be seen at https://www.openup.hk/privacy-policy.htm?lang=en. Help-providers gave their consent to using their text data for research purposes by signing consent forms. The study protocol was approved by the Human Research Ethics Committee of University of Hong Kong (EA1709039).

**Dataset**. Open Up is a free 24/7 online text-based counseling service in Hong Kong designed to provide timely support to youth and young adults experiencing emotional distress[3,17]. Since its inception in 2018 to date (March 2021), 224 counselors and volunteers have provided more than 50,000 sessions. Help-seekers have the option to access the service through the service's website, Facebook, or instant messaging services, including SMS and WhatsApp. At any given time, each counselor can engage with more than one concurrent user. Box 1 shows a fictitious excerpt between a help-seeker and a counselor.

**Defining blocks**. To introduce the counseling session inclusion criteria, we first defined a *block*, which is the basic unit of analysis of this study. Specifically, each session was divided into continuous blocks, each of which contains an arbitrary number of ten messages from the help-seeker. Blocks were classified into three types: ideation about self-harm and suicide blocks (ISSBs), blocks prior to ISSB blocks (PISSBs), and non-ISSB blocks (NISSBs) for the remaining blocks. Using blocks as the basic unit, we transformed the problem of predicting the transition point into the problem of detecting PISSB. Below we define the three different block types in detail.

We first define ISSBs. ISS includes thinking about, considering, or planning self-harm and suicide[18]. It is considered an important risk factor for self-harm and suicide attempts[19]. In this study, the identification of ISS blocks required two steps: Firstly, we relied on C-SSRS rules and keywords-based rules[20–22] to obtain a set of preliminary ISS blocks (*preliminary ISSBs*). Supplementary

---

**Box 1. ▌ An excerpt of a fictitious exchange between a help-seeker and a counselor in English translation. "H" and "C" stand for "help-seeker" and "counselor", respectively. Refer to Supplementary Methods-A for the Chinese version**

H: I am struggling every day to not jump out of the window. I am suffering and I feel so lonely.
C: Hearing this, I am worried about you, Lily
C: It seems that you feel tormented every day.
H: I feel so muddled. What is the point of life?
H: I want to die so badly. I am suffering and I feel so lonely.
C: I am here with you, Lily, you are not alone.
H: I have to wait until October before going back to the clinic for the follow-up consultation. I don't know how to hold on.
H: I want to jump off the building right now. How can I hold on till then?
C: Is it possible to make an earlier appointment with the doctor?
C: Let them know that you are not feeling well lately?
......

**Table 1 Typical false alarms when relying on keywords matching alone.**

| False alarm type | Examples |
| --- | --- |
| Negation-induced: The action or ideation is negated. | 1) It's not that I want to die. |
| | 2) I do not have suicidal thoughts. |
| Subject-induced: The action or ideation is about others rather than about the help-seeker. | 1) They said they really wanted to die, but didn't know how to talk about it. |
| | 2) A friend even died by jumping |
| Tense-induced: The action or ideation happened in the past and not at present. | 1) At the end of June this year, I jumped of a building. |
| | 2) I used to have very severe suicidal tendencies. |
| Other types | syllepsis: |
| | Before, I struggled immensely every time I saw any English (Literal: 'Every time I saw any English I'd want to kill myself') |
| | quoting others: |
| | 1) They said I was only pretending to be suicidal because I was attention seeking. |
| | 2) ...Telling me why not just jump and go kill myself. |
| | dream: |
| | 1) Every time they argue I get nightmares, and in each one I fall off a building |
| | 2) There were several nights where I had nightmares of jumping off a rooftop |

Methods-B reports words/phrases related to the explicit disclosure of ISS used in this study.

It is recognized that there currently is not a set of one-size-fits-all rules to deal with the categorization of ISS. Some rules are relatively loose; others are tighter. Several typical categories of false positives are listed in Table 1 and Table S1 (Supplementary Methods-C). Therefore, after a round of keywords-based coarse selection, we relied on human coders to refine the coding, making sure that they are indeed ISSBs.

The term PISSB is then self-explanatory: It stands for the block immediately prior to an ISSB. All remaining blocks that are neither ISSB nor PISSB were classified as NISSB.

**Session inclusion criteria**. The analysis included only sessions with at least ten message exchanges to provide sufficient information to make the classification. Sessions were excluded if they contained only ISSB and no PISSB (i.e., insufficient content before ISSB) as our aim was to identify PISSB, which was by definition prior to ISSB.

**Network-based PISSB detection**. Below, we first provide a brief description of the general framework before introducing the details of the model.

*General framework*. Words are not scattered randomly in written texts. Instead, they are organized following specific linguistic patterns[23–25]. The syntactical features of texts can be reflected by complex networks, such as word affinity networks (WANs) which depict the semantic relationships (e.g., word adjacency, word similarity) between words. Words occurred adjacently in texts tend to form localized neighborhoods, known as *word modules*[23–25]. Against this backdrop, we first constructed a WAN using a subset of counseling sessions as corpus. Then we examined if words in the divided blocks would form word modules in WAN. If this premise was met, we conducted the third step exploring if comparing the *network-based distance* of PISSB/NISSB word modules with ISSB word modules was a reliable measure of discriminating PISSB from NISSB. The underlying assumption was that closer distance between word modules indicates linguistic proximity between text blocks. Below we describe each step in detail.

*Step 1: Constructing a word affinity network*. According to the overall framework proposed above, we started by constructing a WAN using a subset of sessions, which included sessions conducted in 2019 ($N = 20,618$). Different from English, there is no

obvious boundary for Chinese words. For example, "pine, apple, and elm" in Chinese looks like "pineappleandelm". Therefore, word segmentation was the first step for constructing WAN. Jieba, which is a cutting-edge Chinese word segmentation algorithm, was used to carry out this procedure. Jieba is publically available at https://github.com/fxsjy/jieba.

The second step for constructing WAN was to extract important words to be included in the network. Word filtering is a routine pre-processing step in language modeling because usually a large proportion of words are trivial in any given corpus; they contribute little information and introduce noise to the downstream analyses. We extracted top 5000 important words from the 20,618 sessions by calculating each word's term frequency-inverse document frequency (TF-IDF), a commonly used statistic to measure the importance of a word in the whole corpus. These words constructed the nodes of WAN.

After determining the words to be included in the network, we proceeded to identify meaningful relationships (i.e., links) between them. Links were identified from two aspects: (1) we considered the well-established co-occurrence relationship between words[23]. The co-occurrence of words reflects syntactical features of texts; (2) we also linked up two words in terms of their semantic similarity. In particular, a Word2Vec model using CBOW was trained using these 20,618 sessions. We trained our model with 128 features using the Gensim Python library. By considering two different relationships, *affinity* can be interpreted as a two-fold measure: word co-occurrence and word similarity, the combination of which is more comprehensive than each one alone. A well-designed network is important because it is the fundamental base map for the whole study. A subgraph of WAN is illustrated in Fig. 1a.

*Step 2: Modularity test*. After the WAN was constructed, we proceeded to conduct the word modularity test. Current literature suggests that words in a paragraph are not scattered randomly in WAN; instead, they tend to form localized neighborhoods, known as *word modules*[23,25,26]. Indeed, many real-world networks are modular, characterized by the existence of tightly connected clusters[27]. To investigate if there existed the modular topology of words within a session block, we calculated the average number of steps along the shortest paths for all possible pairs of words in the block (i.e., *mean shortest path length* $l_{block}$) using

$$l_{block} = \frac{\sum_{v_i, v_j \in block} d(v_i, v_j)}{n^2} \quad (1)$$

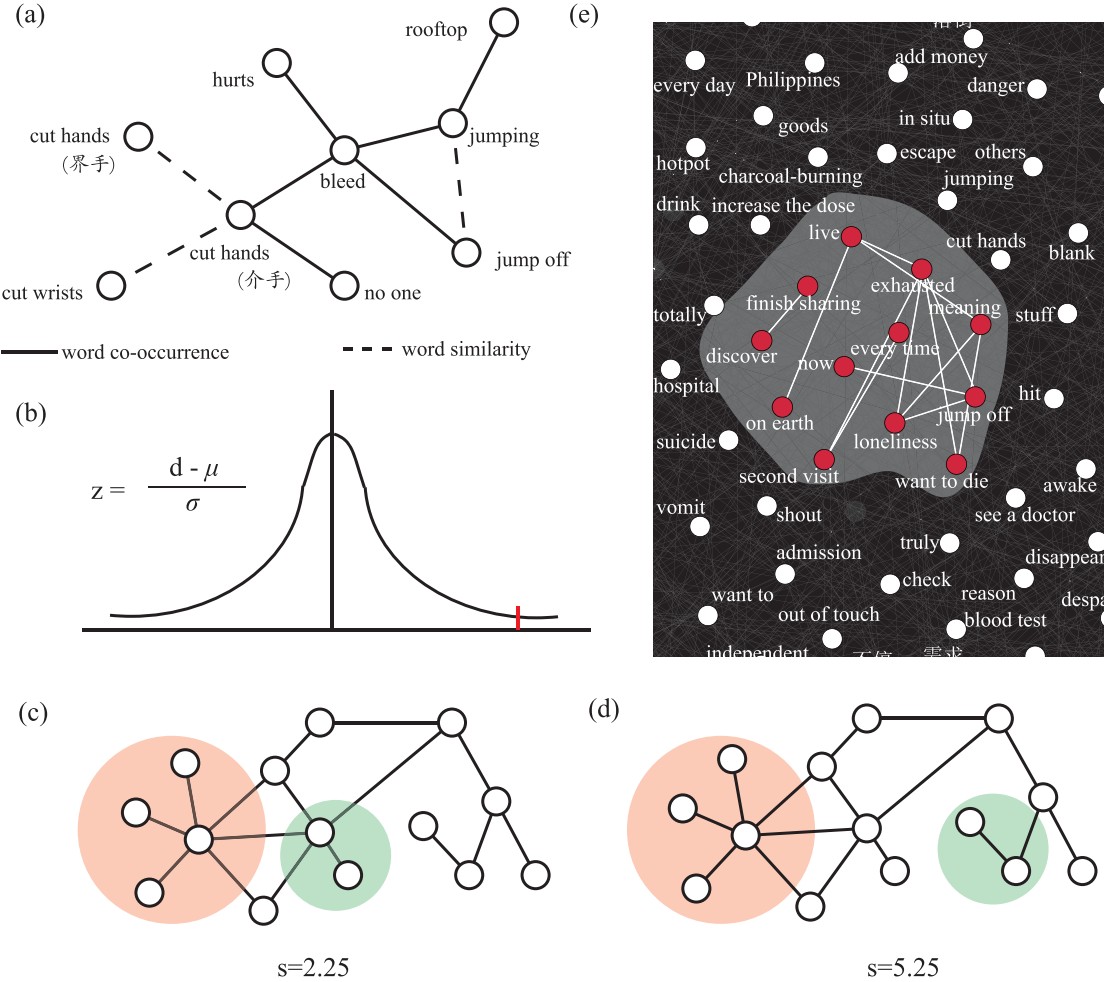

**Fig. 1 The network-based transition point prediction method. a** A subgraph of the word affinity network. Solid lines represent relationships of word co-occurrence, dash lines represent relationships of word similarity. In the rest of the paper, these two types of relationships are no longer visually discriminated for clearer presentation (i.e., solid lines for both types); **b** One-tailed $z$-test; **c**, **d** S-scores for two modules that are topologically closer or topologically distant. **e** Footprints of a block. Words within a block tend to form a connected subgraph, or a *block module*. See Supplementary Methods-D for the bilingual version of **a**, **e**.

where $d(\cdot)$ represents the shortest distance between word $v_i$ and word $v_j$, and $n$ denotes the number of words in the block. Correspondingly, a reference distance is defined as the mean shortest path length for a randomly selected group of words of matching size and degree distribution in WAN. A module appeared if the observed distance $l_{block}$ and the reference distance were statistically different. A one-tailed $z$-test was performed (Fig. 1b) to test the hypothesis that the mean of the average shortest path between words within blocks was shorter than that among random words. Alpha of 5% was selected with a one-tailed test to testing for the possibility of the relationship. Same methods were also adopted in previous literature examining the modular topology of drug-targeted proteins in protein-protein interaction networks[14].

*Step 3: Measuring the distance between blocks.* We then measured the distance between blocks to investigate if such network-based instrument was reliably indicative of detecting PISSBs (i.e., predicting ISSBs). We defined *the distance between two blocks x and y* as the topological distance between two blocks' word modules in WAN, where the topological distance was evaluated using the recently introduced separation measure s-score:[14]

$$s_{(x,y)} = <b_{(x,y)}> - \frac{<l_x> + <l_y>}{2} \qquad (2)$$

where $l_{block}$ is defined in Eq. (1), and $<b_{(x,y)}> = \frac{\sum_{v_i \in x, v_j \in y} d(v_i, v_j)}{m \times n}$ calculates the mean shortest path length between $x$ and $y$. $m$ and $n$ are the number of words in $x$ and $y$, respectively. As such, the s-score compared the mean shortest distance within each block, $<l_x>$ and $<l_y>$, to the mean shortest distance $<b_{(x,y)}>$ between $(x,y)$ block pairs. Topologically, the smaller the s-score is, the closer two blocks are. See Fig. 1c, d for two examples of the calculation of $^S(x,y)$.

Based on the definition of distance between two blocks, we can further calculate the mean distance between two sets of blocks. In particular, the mean distance between a set of PISSBs and a set of ISSBs $\bar{s}_{(PISSBs, ISSBs)}$ was formulated as $\bar{s}_{(PISSBs, ISSBs)} = \frac{\sum_{x \in PISSBs, y \in ISSBs} s_{xy}}{u \times w}$. Similarly, the mean distance between a set of NISSBs and a set of ISSBs $\bar{s}_{(NISSBs, ISSBs)}$ was calculated by the following equation $\bar{s}_{(NISSBs, ISSBs)} = \frac{\sum_{x \in NISSBs, y \in ISSBs} s_{xy}}{v \times w}$. Here, $u, v, w$ denote the number of blocks in PISSB set, NISSB set, and ISSB set, respectively. A $z$-test was performed to examine if the mean values of the two scenarios were statistically different.

*Step 4: Optimal classification threshold.* The last step was to determine the specific cutoff s-score $\bar{s}_{(m, ISSBs)}$ that could be used to evaluate if a block $m$ is a PISSB or not. A recommended approach

for such task is to find the threshold which corresponds to the highest summation of sensitivity (i.e., recall, or true positive rate) and specificity (i.e., true negative rate) for all possible threshold values. Formally, we formulated the PISSB detection problem as a binary classification problem (PISSBs vs. NISSBs). The PISSB set and NISSB set together was randomly split into a training set (80%) and a test set (20%). Given a block $m$ in the test set, we aimed to identify if $m$ was a PISSB or NISSB, by comparing $\bar{s}_{(m, ISSBs)}$ with a threshold $t$, where $\bar{s}_{(m, ISSBs)} = \frac{\sum_{x=m, y \in ISSBs} s_{xy}}{1 \times w}$ by definition (See Eq. (2)), and $t$ was the optimal threshold that maximized sensitivity plus specificity in the training set. If $\bar{s}_{(m, ISSBs)} < t$, $m$ was likely a PISSB. Otherwise, it was classified as an NISSB.

**Statistics and reproducibility.** Data preprocessing and analysis were performed using Python (version 3.6). Network was visualized using Gephi (version 0.9.2). Given its sensitivity, the raw transcript data cannot be made available to the public.

When comparing two groups of interest in this study, one-tailed z-test is used because we aim to testing for the possibility of the relationship in one direction (that said, significantly greater than or significantly less than). A two-tailed test will test both if the mean of $X$ is significantly greater than and if the mean is significantly less than the mean of $Y$. The mean of $X$ is considered significantly different from the mean of $Y$ if the test statistic is in either the top or bottom of its probability distribution, in which case we lose direction.

**Reporting summary.** Further information on research design is available in the Nature Portfolio Reporting Summary linked to this article.

## Results
Based on session inclusion criteria, two years' worth of counseling sessions ($N = 49{,}770$) from Open Up were included in the analysis. Sessions from the year 2019 ($N = 20{,}618$) were used to construct the WAN (i.e., Step 1). Sessions from the year 2020 ($N = 29{,}152$), including 4583 with explicit ISS, were used to train and test the model (i.e., Steps 2, 3, and 4). Among the 4583 sessions, 3415 (74.5%) had ISSBs but no PISSBs. They were excluded from the analysis.

All valid sessions were divided into blocks. In the current analysis, we focused on the first ISSB in each session with ISS. Consequently, 1168 ISSBs and 1168 corresponding PISSBs were extracted from the 2020 dataset. The same number of NISSBs ($N = 1168$) were randomly sampled from the 2020 dataset as counterparts.

**Modularity test.** The resulting WAN from the 2019 dataset contained 5000 words selected by TF-IDF. For the 3504 blocks (i.e., the sum of 1168 ISSBs, 1168 PISSBs, and 1168 NISSBs), the average shortest path length in WAN between words within a conversation block was 2.04 (SD = 0.29), whereas the measure was 2.67 (SD = 0.36) between random words, which was 31% longer ($z = -4.9$, $p < 0.05$). This informs us that, in WAN, words within a block tend to form word modules. An example of a word module derived from a block is shown in Fig. 1e.

**The s-score.** The mean distance between the PISSB set ($N = 1168$) and the ISSB set ($N = 1168$) $\bar{s}_{(PISSBs, ISSBs)}$ was 0.23 (SD = 0.03). The mean distance between the NISSB set ($N = 1168$) and the ISSB set $\bar{s}_{(NISSBs, ISSBs)}$ was 0.30 (SD = 0.05). Figure 2a demonstrates the distribution of the $\bar{s}_{(PISSBs, ISSBs)}$ and $\bar{s}_{(NISSBs, ISSBs)}$. This suggests that network-based distance provides a

potential opportunity to distinguish PISSBs from NISSBs as their distance to ISSBs is statistically different. The z-statistic ($z = -8.8$, $p < 0.05$) suggests that the observed difference between $\bar{s}_{(PISSBs, ISSBs)}$ and $\bar{s}_{(NISSBs, ISSBs)}$ cannot be attributed to chance.

**Optimal decision threshold.** Per Method-Step 4, we set 80% blocks in the PISSB set and the NISSB set ($N = 1668$) as the training set to seek the optimal decision threshold $t$, and used the remaining PISSBs and NISSBs ($N = 668$) to evaluate the performance at $t$. The ROC Curve on the training set is presented in Fig. 2b. It plots the true positive rate (i.e., sensitivity) against the false positive rate (i.e., 1–specificity) for all possible cutoff values. In this study, the threshold was 0.27 (the red circle in Fig. 2b, which corresponds to the highest sensitivity plus specificity). Under this classification threshold, the c-statistic was 0.792 for the training set and 0.787 for the test set, indicating that the performance of using s-score to discover PISSB is good. We replicated the experiment 50 times. In each trial, NISSBs were re-sampled from the counterpart pool. The mean c-statistic for test set was 0.773 (SD = 0.019). This result shows that difference between ISSBs and NISSBs is robust and generalizable. In Fig. 3, we intuitively illustrated a subset of the shortest paths among three randomly sampled blocks in the test set.

## Discussion
In this study, a network-based framework was developed and tested to predict disclosure of self-harm and suicide ideation before its emergence. By doing so, the framework can, in effect, identify users of a text-based online counseling service who may soon disclose ISS in the session. We found that network-based distance between a given block and ISS blocks $\bar{s}_{(m, ISSBs)}$ was a reliable indicator of the likelihood of expressing ISS afterward. The c-statistic was 0.792 for the training set and 0.787 for the test set. This suggests that the language patterns in our study context can accurately predict the emergence of self-harm and suicide ideation. It is noteworthy that existing studies mainly focused on discriminating text with ISS from non-ISS text. What we have done in the present study was to take a step forward by identifying text that would likely precede the disclosure of ideation about self-harm and suicide.

Another contribution of this study is incorporating network science into the solution of NLP tasks. Standard deep learning models, which is the typical way for dealing with NLP problems, are often adequately accurate. Yet they are complex, computationally taxing, and essentially black boxes that are difficult to convince end users. The (fortunate) low base rate in ISS utterances also poses a challenge of trying to find a needle in a haystack because the rare samples are usually not adequate to train the large number of parameters in deep learning models. Alternatively, by making use of complex network theory, in this study, we developed a network-based approach to process language. We demonstrated that network-based framework is simple yet powerful, adding online conversation to a growing list of fields of research that are actively capitalizing on it, such as drug combinations[14], protein–protein interactions[15], disease–disease relationships[16], and drug repurposing for COVID-19[28]. This study serves as an important step to advance suicide risk prediction in the growing field of online psychological services.

The model developed in this study has several important practical implications. Like other synchronous online text-based services, Open Up counselors often need to engage multiple users simultaneously in order to meet the service demand. The proposed method can therefore improve counselors' readiness to provide timely service to those with heightened suicidal risk by alerting them when the linguistic model is reaching a PISSB. This

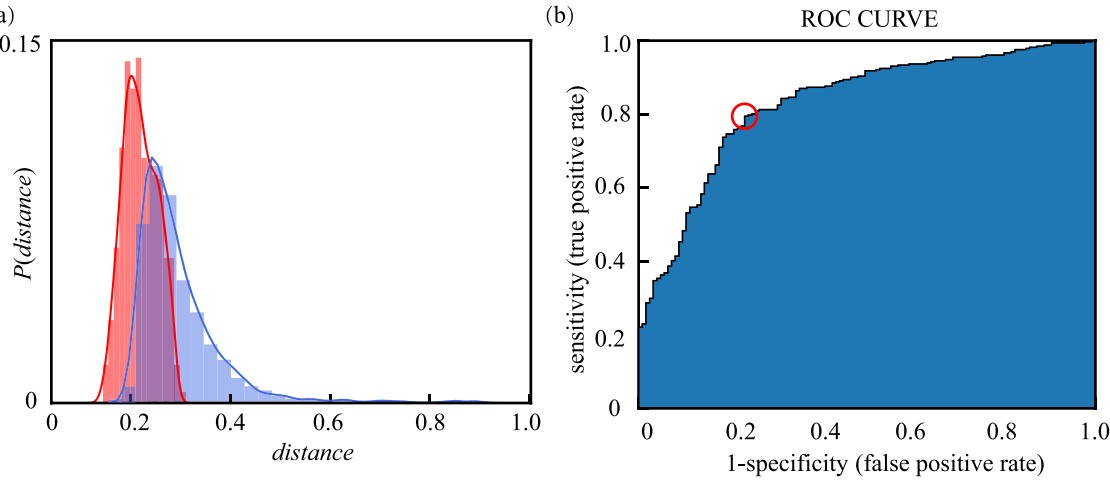

**Fig. 2 Optimal decision threshold. a** The distribution of the $\bar{s}_{(NISSBs,ISSBs)}$ (blue bars) and $\bar{s}_{(PISSBs,ISSBs)}$ (red bars); **b** The ROC-AUC curve for the training set.

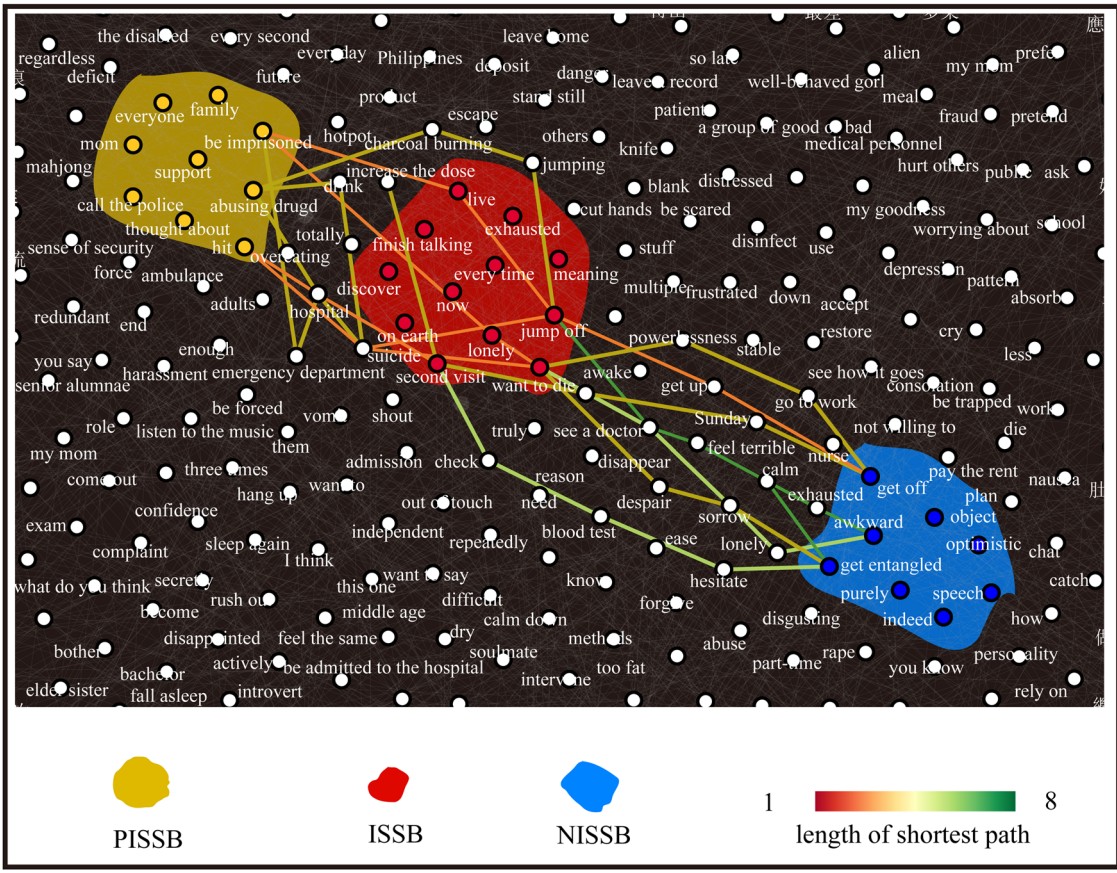

**Fig. 3 Illustration of 18 shortest paths.** We arbitrarily selected three nodes in each of the three randomly sampled block modules which consist of a PISSB module (yellow), a ISSB module (red), and a NISSB module (blue). Paths between the three arbitrary nodes of the PISSB module and the three arbitrary nodes of the ISSB module are shown, resulting in nine paths. Same procedure was conducted for NISSB-ISSB node pairs, resulting in the other nine paths. See Supplementary Notes for the bilingual version of this figure.

real-time alert is critically important and necessary because patients who may soon express suicidal thoughts require the full attention from the counselors. Without this tool, a counselor involved in multiple concurrent chats might be caught off-guard. Monitoring the sessions using the present model could help to improve efficiency and enhance user's experience.

We give an example of how this method may be implemented in the context of online text-based counseling services: A signal can be turned on in the provider's platform if an impending ISS is predicted by the model. Seeing the signal, the counselor can choose to adjust his or her risk assessment/management strategies. After a while (e.g., ten exchanges from the help-seeker), the counselor can re-evaluate if the prediction is accurate: A table showing the five shortest paths appears to guide the counselor in understanding the underlying mechanisms of the model's judgment, such that counselors can quickly understand the logic and then judge the accuracy and usefulness of the model. Table 2 demonstrates an example of a set of such shortest paths with

**Table 2 The shortest paths and their lengths.**

| Five shortest paths | Path lengths |
|---|---|
| abusing drugs—want to die | 0.21 |
| abusing drugs—increasing the dose—want to die | 0.27 |
| be imprisoned—exhausted | 0.31 |
| abusing drugs—drink—suicide | 0.36 |
| abusing drugs—jump off | 0.36 |

respect to the sample conversation in Fig. 3. Path length between two words is calculated using one minus affinity score such that a larger value represents a longer path length. Lastly, by incorporating the feedback from the counselor, the accuracy and usefulness of the model can be improved recurrently.

Second, and relatedly, the implication is that the model can remind counselors, especially new recruits and volunteers, to consider altering their strategies to better elicit disclosure of suicidal ideation as people tend to conceal their suicidal thoughts during psychological support and treatment[11]. An alert flagging a PISSB can help the counselor and their supervisor make adequate adjustments in their services.

Third, the forecasting of forthcoming suicidal ideation by the proposed method can be more sensitive and objective than human because it is not influenced by subjective opinions, which may differ from counselor to counselor. This is especially relevant to new recruits and volunteers who might possess different levels of skills and experience.

Fourth, the current study focused on suicidal ideation, the prevention of which is of the highest priority among many online counseling services. However, we are hopeful that the same method and technology can be applied to predicting the disclosure of other types of risk. Now that the model toward the prediction of suicidal ideation was developed and tested, modifying it with respect to other outcomes of interest is the next step.

Certainly, false positives might be an issue if the proposed model were to be implemented in a wider context. In our 2020 dataset, 15.7% of the sessions were annotated as ISS admissions, which is significantly higher than the general population[29,30]. In such a context, we should pay extra attention to the false positive issue as the base rate of ISS in the general population is likely smaller.

It should be noted that while satisfactory, the accuracy of the model has room for further improvement. The proposed work is a retrospective study that aims to demonstrate the effectiveness of the model. Future research should consider using up-to-date corpus to construct WAN and conduct prospective analysis. It has great promise for model improvement because the latest corpus might involve unseen words and relationships, such as "COVID-19-bankruptcy", etc. On the other hand, and relatedly, the word affinity network should be updated recurrently over time when being deployed to the frontline systems. Also, early detection of conversations with self-harm and suicide ideation should not be equated with the prevention of suicide itself. Because the dataset we used lacks behavioral outcome data, we must acknowledge the limitations that the lack of explicitly expressed suicidal ideation should not be interpreted as the actual lack of such thoughts or plans. Similarly, the expression of suicidal ideation should not be interpreted as imminent suicide risk per se[18]. The model we developed should be treated as merely an additional tool to augment the service provider's risk assessment.

Secondly, the model developed was based on a large-scale Cantonese database. While we are optimistic that the method can be readily applied to other languages or contexts, the generalizability of our findings requires further empirical support. Relatedly, the database was from an online text-based counseling

service; we do not assume the same modules or model are directly applicable to other types of text-based conversations.

Third, we recognized that while what crisis workers said might be as useful as help-seekers in predicting suicidal ideation, the proposed model, however, is not intelligent enough to cope with two tracks of language at the moment. A more sophisticated model (that said, a deep learning model) might deal with the two-phase language patterns to further improve the accuracy, but we might lose explainability.

## Conclusion

Our study demonstrated the ability to predict suicidal ideation prior to its emergence in the context of an online text-based counseling service. By detecting and alerting the help provider before ideation emerges, our model enables the counselors to enhance their preparedness and offer timely intervention. It can potentially also help improve service efficiency through improved triaging and timely deployment of human resources.

## Data availability

Given its sensitivity, the raw transcript data cannot be made available to the public. The source data underlying Fig. 2a, b, 3 can be found at https://doi.org/10.6084/m9.figshare.20292096.

## Code availability

The code used to perform the analysis can be found on GitHub[31]. Data preprocessing and analysis were performed using Python (version 3.6). Network was visualized using Gephi (version 0.9.2).

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

## Acknowledgements

We are grateful for the funding support from the Hong Kong Jockey Club Charities Trust and the Collaborative Research Fund (C7151-20G) for providing the resources to set up Open Up and its operation cost. The support from the strategic theme base research from The University of Hong Kong is gratefully acknowledged.

## Author contributions

Z.X., Q.Z., C.S.C., and P.Y. formulated the idea. Z.X., L.H., F.C., E.C., J.F., and C.T. performed the literature review and annotated the suicidal ideation. Z.X. and Y.X. developed the model and conducted the experiments. Z.X., C.S.C., Q.Z., J.Y., and P.Y. analyzed and interpreted the results and wrote the article. All authors had full access to all data (including statistical reports and tables) in the study and take responsibility for the integrity of the data and the accuracy of the data analysis.

## Competing interests
The authors declare no competing interests.
