## [Peer Review File · Communications Medicine]

Reviewers' comments:

Reviewer #1 (Remarks to the Author):

This is an interesting study that seeks to implement network analysis to generate a model capable of giving advanced warning to crisis line supporters using two way text of an impending transition to the disclosure of suicidal ideation SI. The authors assert that their approach can shed needed light on model explainability. The authors focus on the period prior to suicidal mention, which is critical for the development of suicide prevention tools. The approach is novel and results are interesting and compelling, generating a c-statistic in the range ~80 range, which is pretty good for SI. The statistics appear appropriate; however, please note the analysis suggestion below.

Notably, the strength of the reported prediction efficacy will be dependent on the base rate of the SI in the population; however, in this context, the population refers to individuals writing into the crisis line. Can the authors speak to the proportion of users who express SI on Open Up and the implications of their model for generation of true and false positives in this cohort? I imagine that SI admission would be much higher than the general population, so that generation of false positives (a problem for suicide prediction in general) would be decreased. A note of this in the discussion would strengthen the paper. Furthermore, the authors might speak to whether false positives would even be a problem. In some crisis text lines, for example, suicide risk assessment is always done. In effect then, there would be no danger to false positives but potentially some positive of advanced warning. If it is always done in Open Up, it could be similar. If it is not, could this drive the identification of new cases?

In general for this paper, I have only one analysis suggestion that, if feasible, I feel would strengthen the paper. Beyond that, I have a number of comments about formatting for clarity.

Analysis comment:

Instead of matching NSIBs to the number of SIBS only once, can a permutation analysis of the method, testing randomly selected matched NSIB groups be used to generate a distribution of c statistics on the test set? Would the distribution mean c-statistic be in the range of the identified c-statistic for the study? This analysis approach would add robustness to the interpretation and confidence that the PSIBs can be distinguished using the method from the full range of NSIB possibilities. The alternative is that, for some reason, the authors got lucky with the control set. Why not test it against the bulk of the available data?

Minor comment: Notably, in the definition for SI, terms for self-harm and cutting are included; however, it is noted that self-harm and suicide can represent distinct phenomenon, where folks with self-harm do so to relieve anxiety but have no intention to die.

Minor comment requiring action: I believe the references to supplementary sections I-A and I-B are reversed in the supplement provided to the reviewers. A.) refers to the definition of SI and B.) is the help-seeker/counselor exchange.

Minor comment requiring action: Similar to the above, are the letter labels for Figure 2 correct? 'e' looks like the footprint, 'b', looks like the z-test and 'c-d' look like the s scores. The text refers to Figure 2b as the z-test, but the legend says : "(b) Footprints of a block. Words within a block tend to

form connected subgraphs, called blockmodule". Please correct and make consistent.

Comment not requiring action: This is a random thought prompted by your paper. I wonder if another implication of the explainability of your model is that you might be able to eventually develop a tool to give crisis responders to help draw out SI admission in those reluctant to answer. In other words, could you ask questions about the SI precursors and would this result in higher admissions of SI? In short, instead of observing natural PSIB occurring words, could you drive discussion of PSIB concepts to facilitate admission/mention of SI or truthful answer to SI screening? It's recognized that is well beyond the scope of this study and probably also not going to happen on crisis lines, but it would be interesting. Finally, it would be interesting if the WAN model had efficacy in other text based data such as social media prior to suicidal mention.

Reviewer #2 (Remarks to the Author):

The authors suggested a clear perspective to predict an individual's likelihood of expressing suicidal ideations. Accordingly, the study heuristically categorizes the longitudinal posts of an individual into blocks of sentences with labels: Non-Suicidal Ideation Blocks, Suicidal Ideation Blocks, and Blocks prior to Suicidal Ideation. Because of such categorization, the authors used logistic regression and statistics to describe the approach's applicability quantitatively.

Strengths:

1. The approach to estimating the likelihood of an individual to show signs of suicidal ideations (a terminal mental health condition) from a less difficult or less risky situation is easy to understand.
2. The authors have validated each of the components of the approach and have conducted the experiments in a real-world dataset.
3. As stated by the authors, the factor of explainability is acceptable; however, overly stated.
4. Through statistical means, the authors showed that the proposed approach works.

Weakness:

1. There has been substantial work estimating suicide risk and detecting the factors that cause the drift to suicide risk. It would be interesting if the authors would utilize the guidelines used to train the volunteers in "Open Up" to categorize the posts as SIB, PSIB, or NSIB.
2. Process knowledge like C-SSRS (Columbia Suicide Risk Severity Rating Scale(described in the following two studies (a) and (b)) or SAD PERSONS scale, or https://en.wikipedia.org/wiki/Assessment_of_suicide_risk) could help estimate not only suicidal ideation but other high-risk suicidal behavior.
3. Use of Word Affinity Matrix or Word Modules without the support of background knowledge, though somewhat explainable, would require a huge effort from the clinicians in decision making. It would be nice if the authors discussed how clinicians would use their approach in practice in detail.
4. The example conversation provided by the authors seems a little weird. In the conversation block

on the left side, why does the counselor (C) uses sentences like "Keep feeling lonely and helpless" or "Feeling lack of motivation." It seems the conversational scripts lack empathy and curiosity

5. I like the author's direction on explainability using simple and effective machine learning techniques. It would be good to add a paragraph or two (with examples) on "how the approach can support reasoning or inferencing."

6. Not a major comment, but I urge the authors to proofread the manuscript, make it easy to flow with less convoluted sentences. For example: "Toward these perspectives, our knowledge about predicting suicidal ideation in a text-based online counseling session is still lacking, and a transparent yet robust based online counseling session is still lacking, and a transparent yet robust model for such analysis is needed."

Suggested References:

(a) <https://journals.plos.org/plosone/article?id=10.1371/journal.pone.0250448>

(b) <https://dl.acm.org/doi/pdf/10.1145/3308558.3313698>

Reviewer #3 (Remarks to the Author):

This is a novel and original work that seeks to detect moments of transition before revealing suicidal ideation in consultants to an online synchronous text-based counseling service.

There is originality in the objectives (to specifically capture the moments of transition to unveiling suicidality) and in the methodology by relying on "complex network theory in large scale text-based counseling data" moving away from the deep learning algorithms that are used in other patient text analysis studies.

The methodology of the study includes the construction of a word affinity network by means of semantic affinities, definition of three types of blocks, suicidal ideation blocks (SIBs), prior to suicidal ideation blocks (PSIBs), non-suicidal ideation blocks (NSIBs) and, finally the comparison of "the network-based distance of PSIB/NSIB word modules with SIB word modules".

The authors obtain stimulating results by demonstrating, for example, a statistically significant shorter distance between PSIBs and SIBs.

The authors discuss and conclude about the clinical usefulness of this system in the sense of being able to alert counselors about the moment when suicidal ideation is going to be unveiled. It is very interesting the capacity of the system to capture underlying, hidden aspects of the discourse, prior to the unveiling of suicidal ideas, which implies a significant help of the computational system to the psychological work of human beings.

It is therefore a study of great clinical relevance and great potential for usefulness. To make it more friendly for the clinical reader, I find that it is not clear how they define the contents of the SIB, PSIB, NSIB and it would be desirable, in the final part, to describe in more detail how the system could be implemented in a routine care setting.

First of all, we thank all reviewers for their valuable comments and for the careful reading of our paper.

A quick description of the change of terms: Following Reviewer 1's suggestion, we changed "suicidal ideation (SI)" to "ideation about self-harm and suicide (ISS)" because folks with self-harm ideation might have no intention to die. Correspondingly, SIB, PSIB, and NSIB were changed to ISSB, PISSB, and NISSB, respectively. New terms were used in the revised manuscript and in this response letter.

Response to reviewer 1

1. Notably, the strength of the reported prediction efficacy will be dependent on the base rate of the SI in the population; however, in this context, the population refers to individuals writing into the crisis line. Can the authors speak to the proportion of users who express SI on Open Up and the implications of their model for generation of true and false positives in this cohort? I imagine that SI admission would be much higher than the general population, so that generation of false positives (a problem for suicide prediction in general) would be decreased. A note of this in the discussion would strengthen the paper. Furthermore, the authors might speak to whether false positives would even be a problem. In some crisis text lines, for example, suicide risk assessment is always done. In effect then, there would be no danger to false positives but potentially some positive of advanced warning. If it is always done in Open Up, it could be similar. If it is not, could this drive the identification of new cases?

Authors' response: We thank the reviewer for this comment. During the study period (the year 2020), 4,583 of the 29,152 sessions were annotated as ISS admissions. The proportion of ISS sessions was 15.7%, which is, as suggested by the reviewer, significantly higher than the general population, which is around 6% [1,2], and also a bit higher than those in the age group 18-35 in Hong Kong [3]. We agree with the reviewer that, in such a context, the task of predicting ISS would be easier than in the general population because ISS cases are not extremely rare, and thus the data is not very imbalanced. Although it is not within this paper's scope, it is noteworthy to raise

readers' attention to the false positive issues when applying the proposed model towards the general population, where the base rate of the ISS population is smaller (a problem for ISS prediction in general). Following the reviewer's advice, we add a discussion in the revised manuscript:

“Certainly, false positives might be an issue if the proposed model were to be implemented in a wider context. In our 2020 dataset, 15.7% of the sessions were annotated as ISS admissions, which is significantly higher than the general population [1,2]. In such a context, we should pay extra attention to the false positive issue as the base rate of ISS in the general population is likely smaller.”

2. Instead of matching NSIBs to the number of SIBS only once, can a permutation analysis of the method, testing randomly selected matched NSIB groups be used to generate a distribution of c statistics on the test set? Would the distribution mean c-statistic be in the range of the identified c-statistic for the study? This analysis approach would add robustness to the interpretation and confidence that the PSIBs can be distinguished using the method from the full range of NSIB possibilities. The alternative is that, for some reason, the authors got lucky with the control set. Why not test it against the bulk of the available data?

Authors' response: We thank the reviewer for this important suggestion. Actually, when exploring the model's performance, we indeed ran the code quite a few times using different NISSB samples (because every time we test the code, the NISSB samples would be renewed in a random way.) We should have considered reporting the average performance of the model. In the revised manuscript we add this part:

“We replicated the experiment 50 times. In each trial, NISSBs were re-sampled from the counterpart pool. The mean c-statistic for test set was 0.773 (SD = 0.019). This result shows that difference between ISSBs and NISSBs is robust and generalizable.”

3. Minor comment: Notably, in the definition for SI, terms for self-harm and cutting

are included; however, it is noted that self-harm and suicide can represent distinct phenomenon, where folks with self-harm do so to relieve anxiety but have no intention to die.

Authors' response: This is a very helpful comment! We agree with the reviewer that self-harm behavior does not necessarily indicate suicidal ideation. That said, for the purpose of suicide prevention, the presence of self-harm remains an important risk factor that could increase the risk of suicide. In the revised manuscript, we change the term "suicidal ideation (SI)" to "ideation about self-harm and suicide (ISS)", which we think is a more precise and accurate description.

4. Minor comment requiring action: I believe the references to supplementary sections I-A and I-B are reversed in the supplement provided to the reviewers. A.) refers to the definition of SI and B.) is the help-seeker/counselor exchange. Minor comment requiring action: Similar to the above, are the letter labels for Figure 2 correct? 'e' looks like the footprint, 'b', looks like the z-test and 'c-d' look like the s scores. The text refers to Figure 2b as the z-test, but the legend says: "(b) Footprints of a block. Words within a block tend to form connected subgraphs, called block module". Please correct and make consistent.

Authors' response: We thank the reviewer for pointing these out. We are sorry for the mistake. The paper has been carefully proofread.

5. Comment not requiring action: This is a random thought prompted by your paper. I wonder if another implication of the explainability of your model is that you might be able to eventually develop a tool to give crisis responders to help draw out SI admission in those reluctant to answer. In other words, could you ask questions about the SI precursors and would this result in higher admissions of SI? In short, instead of observing natural PSIB occurring words, could you drive discussion of PSIB concepts to facilitate admission/mention of SI or truthful answer to SI screening? It's recognized that is well beyond the scope of this study and probably also not going to happen on crisis lines, but it would be interesting. Finally, it would be interesting if

the WAN model had efficacy in other text based data such as social media prior to suicidal mention.

Authors' response:

We appreciate the suggestion! We have modified the discussion to allude to this possibility without overstating the implications of our results. We also appreciate the idea of exploring the feasibility of WAN model in other types of platforms. We indeed came across a useful study [4] that aims to discover shifts to suicidal ideation from mental health content in Reddit (similar task, different method and context). So it is indeed interesting and promising to adopt our method in text-based social media and other platforms. We actually are collecting text data from Weibo, a Chinese social media platform. (References for the response are placed at the end of this letter)

Response to reviewer 2

1. There has been substantial work estimating suicide risk and detecting the factors that cause the drift to suicide risk. It would be interesting if the authors would utilize the guidelines used to train the volunteers in "Open Up" to categorize the posts as SIB, PSIB, or NSIB.

Authors' response: We thank the reviewer for this suggestion. Actually, the method the reviewer suggests here is indeed how we adopted in this study to detect ISSB. We are sorry that we didn't make it very clear in the earlier version. In the revised manuscript, we add clear description about how ISSB, PISSB, and NISSB were categorized:

“We first define ISSBs. ISS includes thinking about, considering, or planning self-harm or suicide.¹⁸ Few people who have ISS go on to self-harm or suicide attempts, but ISS are considered an important risk factor for self-harm or suicide attempts¹⁹. In this study, the identification of ISS blocks required two steps: Firstly, we relied on C-SSRS rules and keywords-based rules²⁰⁻²² to obtain a set of preliminary ISS blocks (*preliminary ISSBs*). See Supplementary Section I-A for words/phrases related to the explicit disclosure of ISS used in this study.

It is recognized that there currently is not a set of one-size-fits-all rules to deal

with the categorization of ISS. Some rules are relatively loose; others are tighter. Several typical false positives are listed in Table 1. Therefore, after a round of keywords-based coarse selection, we relied on human coders to refine the coding, making sure that they are indeed ISSBs.

Table 1. Typical false alarms when relying on keywords matching alone.

False alarm type	Examples
Negation-induced : The action or ideation is negated.	a)我唔係想自殺. b)我有自殺念頭.
Subject-induced : The action or ideation is about others rather than about the help-seeker.	a)佢話佢好想死但佢唔知點講好. b)朋友仲要跳樓
Tense-induced : The action or ideation happened in the past and not at present.	a)今年 6 月尾, 我跳左樓. b)我之前一直有好嚴重嘅自殺傾向.
Other types	syllipsis: a)以前見到英文想死.
	quoting others: a)話我扮嘢自殺係 attention seeking. b)叫我點解唔去跳樓死左佢.
	dream: a)佢地次次噪完個晚都會發惡夢但係次次都會跌落樓. b)有幾晚發惡夢自己系天台跳落去.

The term PISSB is then self-explanatory: It stands for the block immediately prior to an ISSB. All remaining blocks that are neither ISSB nor PISSB were classified as NISSB.”

Supplementary Section I-A:

“Words/phrases used to find *preliminary ISSBs* include 想死 (want to die), 自殺 (suicide), 跳樓 (jump), 離開世界 (escape from the world), 死咗 (die), 遺書 (suicide note), 跳落去 (jump from height), 安樂死 (euthanasia), 尋死 (seek

opportunity to die), 去死 (die), 介手 (cut), 界手 (cut), 界刀 (cut), 不想活 (do not wish to live any more), 割脈 (cut), 跳樓 (jump from the rooftop), 快 D 死 (die quickly), 快 d 死 (die quickly), 自刎 (cut one's throat), 天台 (rooftop), 跌落 (fall off), 企跳 (jump), 自殘 (self-harm), and 鐮 (cut).”

2. Process knowledge like C-SSRS (Columbia Suicide Risk Severity Rating Scale(described in the following two studies (a) and (b)) or SAD PERSONS scale, or https://en.wikipedia.org/wiki/Assessment_of_suicide_risk) could help estimate not only suicidal ideation but other high-risk suicidal behavior.

Suggested References:

(a) <https://journals.plos.org/plosone/article?id=10.1371/journal.pone.0250448>

(b) <https://dl.acm.org/doi/pdf/10.1145/3308558.3313698>

Authors' response: We agree with the reviewer that there currently isn't a set of one-fit-all rules to deal with the categorization of self-harm and suicide ideation. Some rules are relatively loose; others are tighter. Therefore, in our study, after a round of keywords-based coarse selection, we invited human coders to help verify and refine our rules. It is recognized that this coding process is labor-intensive. In the context of this study, however, accurately identifying ISSB sessions was our top priority. We added the two references [7,8] the reviewer suggested in the revised manuscript. Please refer to our response to Comment 1 for details.

3. Use of Word Affinity Matrix or Word Modules without the support of background knowledge, though somewhat explainable, would require a huge effort from the clinicians in decision making. It would be nice if the authors discussed how clinicians would use their approach in practice in detail.

Authors' response: We appreciate this very useful suggestion! In the revised paper, we summarized how the proposed model can be implemented in a routine counseling setting:

“We give an example of how this method may be implemented in the context of online text-based counseling services: A signal can be turned on in the provider’s platform if an impending ISS is predicted by the model. Seeing the signal, the counselor can choose to adjust his or her risk assessment/management strategies. After a while (e.g., ten exchanges from the help-seeker), the counselor can re-evaluate if the prediction is accurate: A table showing the five shortest paths appears to guide the counselor in understanding the underlying mechanisms of the model’s judgement, such that counselors can quickly understand the logic and then judge the accuracy and usefulness of the model. Table 2 demonstrates an example of a set of such shortest paths with respect to the sample conversation in Figure 4. Path length between two words is calculated using one minus affinity score such that a larger value represents a longer path length. Lastly, by incorporating the feedback from the counselor, the accuracy and usefulness of the model can be improved recurrently.

Table 2. The shortest paths and their affinity scores.

Five shortest paths	Path length
abusing drugs -- want to die	0.21
abusing drugs -- increasing the dose -- want to die	0.27
be imprisoned -- exhausted	0.31
abusing drugs -- drink -- suicide	0.36
abusing drugs -- jump off	0.36

We are also working on converting our algorithm to a real-time online system and deploying it in Open Up, which can be conveniently used by frontline personals. An earlier version of such system developed by our group, which is designed towards

dealing with Disease Affinity Networks (DAN, similar to Word Affinity Networks (WAN)), can be found here: <http://101.32.178.120:8000/self-harm/>?

4. The example conversation provided by the authors seems a little weird. In the conversation block on the left side, why does the counselor (C) uses sentences like "Keep feeling lonely and helpless" or "Feeling lack of motivation." It seems the conversational scripts lack empathy and curiosity

Authors' response: We thank the reviewer very much for pointing this out. C (counselor) should be H (help-seeker) for the two weird sentences the reviewer mentioned. We are sorry for the typos. The revised manuscript has been thoroughly proofread.

5. I like the author's direction on explainability using simple and effective machine learning techniques. It would be good to add a paragraph or two (with examples) on "how the approach can support reasoning or inferencing."

Authors' response: We thank the reviewer for raising this point. Please refer to our response to comment 3 for details.

6. Not a major comment, but I urge the authors to proofread the manuscript, make it easy to flow with less convoluted sentences. For example: "Toward these perspectives, our knowledge about predicting suicidal ideation in a text-based online counseling session is still lacking, and a transparent yet robust based online counseling session is still lacking, and a transparent yet robust model for such analysis is needed."

Authors' response: Thank you very much for pointing this out! The paper has been proofread.

Response to reviewer 3

To make it more friendly for the clinical reader, I find that it is not clear how they define the contents of the SIB, PSIB, NSIB and it would be desirable, in the final part, to describe in more detail how the system could be implemented in a routine care setting.

Authors' response: In response to the two comments, in the revised manuscript we a) added a clearer description about how we defined the content of the three types of conversation blocks, and b) described how the proposed system could be used in a routine counseling setting:

For comment a):

“We first define ISSBs. ISS includes thinking about, considering, or planning self-harm or suicide.¹⁸ Few people who have ISS go on to self-harm or suicide attempts, but ISS are considered an important risk factor for self-harm or suicide attempts¹⁹. In this study, the identification of ISS blocks required two steps: Firstly, we relied on C-SSRS rules and keywords-based rules^{20–22} to obtain a set of preliminary ISS blocks (*preliminary ISSBs*). See Supplementary Section I-A for words/phrases related to the explicit disclosure of ISS used in this study.

It is recognized that there currently is not a set of one-size-fits-all rules to deal with the categorization of ISS. Some rules are relatively loose; others are tighter. Several typical false positives are listed in Table 1. Therefore, after a round of keywords-based coarse selection, we relied on human coders to refine the coding, making sure that they are indeed ISSBs.

Table 1. Typical false alarms when relying on keywords matching alone.

False alarm type	Examples
Negation-induced: The action or ideation is negated.	a)我唔係想自殺. b)我有自殺念頭.
Subject-induced: The action or ideation is about others rather than about the help-seeker.	a)佢話佢好想死但佢唔知點講好. b)朋友仲要跳樓
Tense-induced: The action or ideation happened in the past and	a)今年 6 月尾, 我跳左樓. b)我之前一直有好嚴重嘅自殺

not at present.	傾向.
Other types	syllepsis: a)以前見到英文想死.
	quoting others: a)話我扮曬嘢自殺係 attention seeking. b)叫我點解唔去跳樓死左佢.
	dream: a)佢地次次噪完個晚都會發惡夢但係次次都會跌落樓. b)有幾晚發惡夢自己系天台跳落去.

The term PISSB is then self-explanatory: It stands for the block immediately prior to an ISSB. All remaining blocks that are neither ISSB nor PISSB were classified as NISSB.”

Supplementary Section I-A:

“Words/phrases used to find *preliminary ISSBs* include 想死 (want to die), 自殺 (suicide), 跳樓 (jump), 離開世界 (escape from the world), 死咗 (die), 遺書 (suicide note), 跳落去 (jump from height), 安樂死 (euthanasia), 尋死 (seek opportunity to die), 去死 (die), 介手 (cut), 界手 (cut), 界刀 (cut), 不想活 (do not wish to live any more), 割脈 (cut), 跳樓 (jump from the rooftop), 快 D 死 (die quickly), 快 d 死 (die quickly), 自刎 (cut one’s throat), 天台 (rooftop), 跌落 (fall off), 企跳 (jump), 自殘 (self-harm), and 鋸 (cut).”

For comment b):

In the revised paper, we summarized how the proposed model can be implemented in a routine counseling setting:

“We give an example of how this method may be implemented in the context of online text-based counseling services: A signal can be turned on in the provider’s platform if

an impending ISS is predicted by the model. Seeing the signal, the counselor can choose to adjust his or her risk assessment/management strategies. After a while (e.g., ten exchanges from the help-seeker), the counselor can re-evaluate if the prediction is accurate: A table showing the five shortest paths appears to guide the counselor in understanding the underlying mechanisms of the model's judgement, such that counselors can quickly understand the logic and then judge the accuracy and usefulness of the model. Table 2 demonstrates an example of a set of such shortest paths with respect to the sample conversation in Figure 4. Path length between two words is calculated using one minus affinity score such that a larger value represents a longer path length. Lastly, by incorporating the feedback from the counselor, the accuracy and usefulness of the model can be improved recurrently.

Table 2. The shortest paths and their affinity scores.

Five shortest paths	Path lengths
abusing drugs -- want to die	0.21
abusing drugs -- increasing the dose -- want to die	0.27
be imprisoned -- exhausted	0.31
abusing drugs -- drink -- suicide	0.36
abusing drugs -- jump off	0.36

We are also working on converting our algorithm to a real-time online system and deploying it in Open Up, which can be conveniently used by frontline personals. An earlier version of such system developed by our group, which is designed towards dealing with Disease Affinity Networks (DAN, similar to Word Affinity Networks (WAN)), can be found here: <http://101.32.178.120:8000/self-harm/>?

Once again, we would like to thank all reviewers for helping us improve the quality of this paper!

References

- [1] Harmer B, Lee S, Duong TvH SA (2021) Suicidal Ideation. *StatPearls Publ.*
- [2] Dalglish SL, Melchior M, Younes N, Surkan PJ (2015) Work characteristics and suicidal ideation in young adults in France. *Soc. Psychiatry Psychiatr. Epidemiol.* **50**, 613–620.
- [3] Fong TCT, Cheng Q, Yip PSF (2021) Change in suicidal ideation and associated factors among young adults in Hong Kong from 2018 to 2019: a latent transition analysis. *Soc. Psychiatry Psychiatr. Epidemiol.*
- [4] De Choudhury M, Kiciman E, Dredze M, Coppersmith G, Kumar M (2016) Discovering shifts to suicidal ideation from mental health content in social media. *Conf. Hum. Factors Comput. Syst. - Proc.* 2098–2110.
- [5] Klonsky ED, May AM, Saffer BY (2016) Suicide, Suicide Attempts, and Suicidal Ideation. *Annu. Rev. Clin. Psychol.* **12**, 307–330.
- [6] GLIATTO MF (1999) Evaluation and treatment of patients with suicidal ideation. *Am. Fam. Physician* **59**, 1500–1506.
- [7] Gaur M, Aribandi V, Alambo A, Kursuncui U, Thirunarayani K, Beichi J, Pathak J, Sheth A (2021) Characterization of time-variant and timeinvariant assessment of suicidality on Reddit using C-SSRS. *PLoS One* **16**, 1–21.
- [8] Gaur M, Kursuncu U, Sheth A, Alambo A, Thirunarayan K, Welton RS, Sain JP, Kavuluru R, Pathak J (2019) Knowledge-aware assessment of severity of suicide risk for early intervention. *Web Conf. 2019 - Proc. World Wide Web Conf. WWW 2019* 514–525.
- [9] Sawhney R, Manchanda P, Mathur P, Shah R, Singh R (2019) Exploring and Learning Suicidal Ideation Connotations on Social Media with Deep Learning. 167–175.

Reviewers' comments:

Reviewer #1 (Remarks to the Author):

Thank you for addressing my comments, especially presentation of the average AUC demonstrating robustness. I am satisfied with the changes and feel the paper is interesting and adds to the literature. Changes made in light of the other reviewer comments also strengthen the paper. I recommend publishing.

Reviewer #4 (Remarks to the Author):

This is a very interesting project trying to identify precursors of disclosure of suicidal ideation. One of the motivators is that crisis workers have more than one person on the line at once, and prioritization would be helpful. I am not an expert on machine learning or natural language processing, although I work with people who are experts, so I will focus on the aspect of the article for which I have some expertise, namely phenomenology of suicidal behavior, assessment of risk, and clinical implications. I have a couple of methodological questions, too, that I will start with:

1. Why are 1-tailed statistical tests used?
2. on line 165, the article states that "there is no obvious boundary for Chinese.." I don't understand this, and it may be that others will not either-- not sure if this has to do with lack of expertise in Chinese, NLP, or both, but there will be lots of readers who lack experience in these important topics.
3. The development of an algorithm in 2019 and validation in 2020 on the surface is a good approach, but I am wondering if there are any implications of the impact of the pandemic on word choice, etc. that might alter word frequency and network relationships.
4. It was not clear to me but it seems the focus was exclusively on what the caller was saying. That is okay, but if so, it is a significant limitation, because it will also be important to know what the crisis worker may have said or not said that led to disclosure. If a crisis worker inadvertently said things that might shut down the caller, then the response of the crisis worker could moderate the relationship between precursors of suicidal ideation and a suicidal ideation outcome. Conversely, learning what a person could say as a crisis worker to encourage disclosure would be an important contribution, if not in this paper, then in the next.
5. One perhaps relevant article to cite is the work of Dr. Mumnun De Choudhury, who, I believe in 2017 examined Reddit posts of individuals who were purveyors of mental health sites; Dr. de Choudhury examined the word choice of those individuals who "migrated" from mental health, to suicide specific sites, and found some interesting precursors.

First of all, we thank all reviewers for their valuable comments which significantly improve the quality of this paper.

Response to reviewer 1

Reviewer #1 (Remarks to the Author):

Thank you for addressing my comments, especially presentation of the average AUC demonstrating robustness. I am satisfied with the changes and feel the paper is interesting and adds to the literature. Changes made in light of the other reviewer comments also strengthen the paper. I recommend publishing.

Authors' response: We thank the reviewer for his/her positive feedback!

Response to reviewer 4

Reviewer #4 (Remarks to the Author):

This is a very interesting project trying to identify precursors of disclosure of suicidal ideation. One of the motivators is that crisis workers have more than one person on the line at once, and prioritization would be helpful. I am not an expert on machine learning or natural language processing, although I work with people who are experts, so I will focus on the aspect of the article for which I have some expertise, namely phenomenology of suicidal behavior, assessment of risk, and clinical implications. I have a couple of methodological questions, too, that I will start with:

1. Why are 1-tailed statistical tests used?

Authors' response: We thank the reviewer for this comment. 1-tailed z -test is used because we aim to testing for the possibility of the relationship *in one direction* (that said, significantly greater than or significantly less than). A 2-tailed test will test both if the mean of X is significantly greater than and if the mean is significantly less than the mean of Y. The mean of X is considered significantly different from the mean of Y if the test statistic is in **either the top or bottom** of its probability distribution, in which case we lose direction. We added those discussions to the revised manuscript:

“A z -test was performed (Figure 2(b)) to test the hypothesis that the mean of the average

shortest path between words within blocks was shorter than that among random words. Alpha of 5% was selected with a one-tailed test to testing for the possibility of the relationship in one direction.”

2. On line 165, the article states that "there is no obvious boundary for Chinese.." I don't understand this, and it may be that others will not either-- not sure if this has to do with lack of expertise in Chinese, NLP, or both, but there will be lots of readers who lack experience in these important topics.

Authors' response: Thank you so much for pointing this out. To exemplify, “pine, apple, and elm” in Chinese looks like “pineappleandelm”, with no space among words. So typically, the first step to cope with Chinese in a NLP task is to use algorithms to explore (a) where to separate characters, and (b) whether “pineapple” or “pine, apple” is the best solution according to the context. Usually, we use a probabilistic model called Hidden Markov Model to deal with such task. We elaborate this point in the revised manuscript, which is also copied below:

“Different from English, there is no obvious boundary for Chinese words. For example, “pine, apple, and elm” in Chinese looks like “pineappleandelm”. Therefore, word segmentation was the first step for constructing WAN.”

3. The development of an algorithm in 2019 and validation in 2020 on the surface is a good approach, but I am wondering if there are any implications of the impact of the pandemic on word choice, etc. that might alter word frequency and network relationships.

Authors' response: We thank the reviewer for this important comment. We conducted additional analysis using data from January 2019-June 2020 to construct the word affinity network and apply it to data from July 2020-December 2020. We observed that c-statistic was improved from 0.773 to 0.791. While such comparison is not exactly rigorous because we test on different research sample size, this result to some extent supports reviewer's hypothesis that there is an impact of pandemic related words on the prediction: New words and relationships were included in this new network, thus

providing new salient information about the pandemic words and suicidal ideation. This inspires us that we should consider update the word affinity network recursively using up-to-date corpus to involve latest semantics when deploying our model to the frontline system. Those discussions have been added to the revised literature:

“It should be noted that while satisfactory, the accuracy of the model has room for further improvement. The proposed work is a retrospective study that aims to demonstrate the effectiveness of the model. Future research should consider using up-to-date corpus to construct WAN and conduct prospective analyses. It has great promise for model improvement because the latest corpus might involve unseen words and relationships, such as “COVID-19--bankruptcy”, etc. On the other hand, and relatedly, word affinity network should be updated recursively over time when being deployed to the frontline systems.”

4. It was not clear to me but it seems the focus was exclusively on what the caller was saying. That is okay, but if so, it is a significant limitation, because it will also be important to know what the crisis worker may have said or not said that led to disclosure. If a crisis worker inadvertently said things that might shut down the caller, then the response of the crisis worker could moderate the relationship between precursors of suicidal ideation and a suicidal ideation outcome. Conversely, learning what a person could say as a crisis worker to encourage disclosure would be an important contribution, if not in this paper, then in the next.

Authors' response: We thank the reviewer for raising this important view! We really resonate with the reviewer about this point. According to literature ¹, crisis workers in OpenUp were trained to encourage disclosure because users tend to conceal their true negative thoughts, including suicidal ideation. What crisis workers inadvertently said might also encourage or shut down the user. Undoubtedly, incorporating what crisis workers said to improve model performance has great promise.

In fact, during experimental setup, we had thought about this point and tried both schemes. Counterintuitively, we found that using words from help-seeker alone achieved a better performance. A possible explanation would be that the language

patterns of help seekers and crisis workers are different. The proposed model is not intelligent enough to cope with two tracks of language. A more sophisticated model (that said, a deep learning model) might deal with the two-phase language pattern issue, but we lose explainability. In the current paper, we sacrifice some accuracy to gain model transparency because in the health care context, transparency is as important as accuracy.

5. One perhaps relevant article to cite is the work of Dr. Mumnun De Choudhury, who, I believe in 2017 examined Reddit posts of individuals who were purveyors of mental health sites; Dr. de Choudhury examined the word choice of those individuals who "migrated" from mental health, to suicide specific sites, and found some interesting precursors.

Authors' response: We thank the reviewer very much for recommending this representative paper ². We actually had noticed it and had already cited it.

Once again, we would like to thank all reviewers and editors for helping us improve the quality of this paper!

References

1. Blanchard M, Farber BA. "It is never okay to talk about suicide": Patients' reasons for concealing suicidal ideation in psychotherapy. *Psychother Res.* 2020;30(1):124-136. doi:10.1080/10503307.2018.1543977
2. De Choudhury M, Kiciman E, Dredze M, Coppersmith G, Kumar M. Discovering shifts to suicidal ideation from mental health content in social media. *Conf Hum Factors Comput Syst - Proc.* 2016:2098-2110. doi:10.1145/2858036.2858207

REVIEWERS' COMMENTS:

Reviewer #4 (Remarks to the Author):

The authors responded thoughtfully to the critique. Personally, I am not convinced that a 1-sided statistical test is ever indicated, but I will leave that decision to the editor and statisticians.

The most substantial concern that I had is that the responses of the crisis workers were not included in the model. The authors responded that they had done a model using crisis workers responses that actually performed better than this one and that integrating the two streams of data was too complex for now.

When thinking about clinical utility wouldn't the crisis workers' knowing what to say be at least as useful as just monitoring the callers. Obviously, both are important, but i think this should be noted as a limitation and that in the future (assuming the authors want to do this), they will report on the crisis workers' responses as predictors.

Manuscript Title: Network-based prediction of the disclosure of ideation about self-harm and suicide in online counseling sessions

We would like to thank the reviewers again for their constructive comments! To respond to the latest appraisals, we elaborated on why one-tailed z-test was adopted instead of two-tailed in the revised manuscript; We also added the limitation of the manuscript with respect to the absence of the crisis workers' responses as indicated by the reviewer. Details can be seen below:

1. Methods---Step 2: Modularity test--- “A z-test was performed (Figure 2(b)) to test the hypothesis that the mean of the average shortest path between words within blocks was shorter than that among random words. Alpha of 5% was selected with a one-tailed test to testing for the possibility of the relationship. “

Statistics and reproducibility --- “One-tailed z-test is used because we aim to testing for the possibility of the relationship *in one direction* (that said, significantly greater than or significantly less than). A two-tailed test will test both if the mean of X is significantly greater than and if the mean is significantly less than the mean of Y . The mean of X is considered significantly different from the mean of Y if the test statistic is in either the top or bottom of its probability distribution, in which case we lose direction.”

2. Discussion --- “Third, we recognized that while what crisis workers said might be as useful as help-seekers in predicting suicidal ideation, the proposed model, however, is not intelligent enough to cope with two tracks of language at the moment. A more sophisticated model (that said, a deep learning model) might deal with the two-phase language patterns to further improve the accuracy, but we might lose explainability.”

Thank you very much!